# Causal factors affecting gross motor function in children diagnosed with cerebral palsy

**Bruce A. MacWilliams**[1,2]\*, **Sarada Prasad**[1], **Amy L. Shuckra**[1], **Michael H. Schwartz**[3,4]

**1** Shriners Hospitals for Children, Salt Lake City, Utah, United States of America, **2** Department of Orthopedic Surgery, University of Utah, Salt Lake City, Utah, United States of America, **3** James R. Gage Center for Gait and Motion Analysis, Gillette Children's Specialty Healthcare, St. Paul, Minnesota, United States of America, **4** Department of Orthopedic Surgery, University of Minnesota, Minneapolis, Minnesota, United States of America

\* bmacwilliams@shrinenet.org

## Abstract

### Background

Cerebral palsy (CP) is a complex neuromuscular condition that may negatively influence gross motor function. Children diagnosed with CP often exhibit spasticity, weakness, reduced motor control, contracture, and bony malalignment. Despite many previous association studies, the causal impact of these impairments on motor function is unknown.

### Aim

In this study, we proposed a causal model which estimated the effects of common impairments on motor function in children with spastic CP as measured by the 66-item Gross Motor Function Measure (GMFM-66). We estimated both direct and total effect sizes of all included variables using linear regression based on covariate adjustment sets implied by the minimally sufficient adjustment sets. In addition, we estimated bivariate effect sizes of all measures for comparison.

### Method

We retrospectively evaluated 300 consecutive subjects with spastic cerebral palsy who underwent routine clinical gait analysis. Model data included standard information collected during this analysis.

### Results

The largest causal effect sizes, as measured by standardized regression coefficients, were found for selective voluntary motor control and dynamic motor control, followed by strength, then gait deviations. In contrast, common treatment targets, such as spasticity and orthopedic deformity, had relatively small effects. Effect sizes estimated from bivariate models, which cannot appropriately adjust for other causal factors, substantially overestimated the total effect of spasticity, strength, and orthopedic deformity.

**Data Availability Statement:** All relevant data are within the manuscript and its Supporting Information files.

**Funding:** The author(s) received no specific funding for this work.

**Competing interests:** The authors have declared that no competing interests exist.

## Interpretation

Understanding the effects of impairments on gross motor function will allow clinicians to direct treatments at those impairments with the greatest potential to influence gross motor function and provide realistic expectations of the anticipated changes.

## Introduction

### Gross motor function in cerebral palsy

Gross motor function in children directly influences quality of life indicators such as activity and participation [1–3]. Children diagnosed with cerebral palsy (CP) often have limitations in their gross motor abilities [4]. An important goal of treatment for these children is to optimize functional abilities. However, knowing what to treat is not obvious, since the causal link from neurological and musculoskeletal factors to gross motor skills is poorly understood.

Cerebral palsy is a complex neurological condition and the most common childhood motor disability with incidence ranging from 2–3.5 per 1000 children [5]. CP describes a group of permanent movement and posture disorders in which damage to the developing brain impairs motor control and may induce spasticity, leading to reduced strength and abnormal musculoskeletal loading, in turn leading to joint contractures and bony deformities. Although the injury is non-progressive, secondary effects can continue to worsen with maturation. Function in this population is commonly measured using the Gross Motor Function Measure (GMFM), a task-based standardized test consisting of five domains that rate a patient's level of function (A. Lying and Rolling, B. Sitting, C. Crawling and Kneeling, D. Standing, and E. Walking, Running and Jumping) [6]. The GMFM may be reported using the full 88-item score (GMFM-88), a 66-item score (GMFM-66) [7], or individual domain scores. The GMFM has been shown to be related to measures of mobility, self-care, and social function [8, 9]. For ambulatory children, the standing (GMFM-D) and walking, running, jumping (GMFM-E) domain scores are highly relevant to mobility.

Treatment options for children with CP span many disciplines including physical therapy, orthopaedic surgery, neurology, and physiatry. Physical therapy treatment focuses on goal oriented task to improve function and participation with interventions typically focused on strengthening, goal directed functional training, treadmill training, casting and bracing. Surgical treatment to address structural changes secondary to motor disorders commonly include lengthening of musculotendon units to improve range of motion, and osteotomy or guided growth to improve alignments. Multiple procedures are commonly performed at one surgical event, deemed single event multilevel surgery or SEMLS. Spasticity management may be attempted with either focal (chemodenervation) or global (baclofen, selective dorsal rhizotomy) approaches [10].

There is little evidence that any of these common treatments for the CP child improve function. Significant improvements in gross motor function have not been found with SEMLS [11–13]. Selective dorsal rhizotomy may result in small functional increases at short- and midterm, but these are not maintained at long term follow-up [14–19]. Likewise, evidence of functional gains from physical therapy treatments are limited to short-term changes [20–22]. Knowing the relative influences of impairments on gross motor function can allow clinicians to target treatments at those impairments with the greatest impact on this important outcome domain.

## Association studies of function

Much of the literature investigating the relationships between impairments and function consists of studies that consider the associations of a few chosen variables (*e.g.*, strength of a single joint). Several such studies investigated relationships between impairments and gross motor function using bivariate correlation or regression; analysis of one independent variable of cause (*e.g.*, spasticity, strength) to one dependent variable of effect (*e.g.*, gait, GMFM) [23–26]. This approach is insufficient when complex relationships between variables exist, since it ignores the dependent effects of other influential variables. These studies focus on associations, with little or no formal discussion of causal mechanisms. For example, Shin *et al.* evaluated bivariate correlations between isometric hip and knee strength in 24 pediatric subjects with CP and GMFM-D, and GMFM-E. Correlations with GMFM domain scores ranged from $r = 0.06$–$0.30$, and none were significant. Other primary (*e.g.*, motor control) and secondary (*e.g.*, contracture) impairments were not considered, and each strength measure was considered in isolation [25].

When multivariate relationships have been explored, it has been done with limited variable sets, small numbers of participants, and without explicitly proposing and testing causal relationships. As an example, Eek and Beckung examined the relationships between hip, knee, and ankle strength and function in 55 children with CP. No other variables were considered. In contrast to Shin, significant bivariate correlations were found ranging from $r = 0.59$–$0.80$ between strength and GMFM-66. The multivariate combination most strongly correlated to GMFM-66 was ankle plantarflexor and hip flexor strength, yielding an $r^2 = 0.73$ [27].

While previous studies identified many factors *associated* with gross motor function, the utility of these models for guiding intervention is limited since pernicious effects like confounding, mediation, and opening of non-causal pathways is possible [28]. By analyzing strength in isolation, without consideration for the causal influences of other variables, Eek and Beckung's results suggest that just two muscle groups account for 73% of the variance in GMFM-66. On the surface, these results imply strengthening these two muscles would be a highly effective therapy. However, neurological impairments do not occur in isolation. If weakness is influenced by poor motor control, age, or overall severity of injury, the effects of strengthening might be smaller than expected. In what follows we show that proper accounting for covariates reduces the apparent importance implied by non-causal analysis of many individual clinical factors.

## Explicit causal approach

Behind most, if not all, association studies lurks an implicit causal hypothesis. It is unlikely that Eek and Beckung were simply trying to demonstrate an association between strength and GMFM-66. Rather, their study almost certainly reflects an assumption that strength is an important *cause* of function. We demonstrate below that strength *is* an important causal factor, though with a substantially smaller impact than bivariate estimates would suggest. However, if causal claims are going to be made, they should be explicit and they should be tested.

An explicit causal model provides a set of conditions that assess the plausibility of the model. Once plausibility is established, a casual model can be used to derive covariate adjustments necessary for statistical modeling. Both the direct effect sizes (dependent variable changes occurring from isolated independent variable changes) and total effect sizes (dependent variable changes occurring when the independent variable is transmitted through mediator variables) can be evaluated to assess the relative influences of model variables [29]. There has been limited use of causal modeling in the study of CP. Kim and Park proposed a simple causal model to examine the effects of spasticity and strength on GMFM-88 and functional

outcomes as measured by the Pediatric Evaluation of Disability Inventory (PEDI) [30]. Only GMFM-88 had significant direct effects on PEDI, whereas spasticity and strength had indirect effects, mediated by GMFM-88. The authors noted that the model lacked other potentially important variables such as orthopedic deformities.

Instrumented three-dimensional gait analysis is a common diagnostic and treatment-planning tool used in the assessment of children with CP. Standard gait analysis protocols include direct and indirect measures of neurological and orthopedic impairments that may influence function. The aim of this study is to hypothesize and test a causal model for GMFM-66 that examines the influence of key neurological and orthopedic impairments commonly measured during a gait analysis visit, which includes a comprehensive physical exam, standard functional measures, and 3D kinematics and electromyography collected during walking. Compared to Kim and Park's study, our proposed model considers measures encompassing all International Classification of Functioning, Disability and Health (ICF, www.asha.org/slp/icf) domains, including measures of orthopedic deformity, spasticity, gait quality, and both dynamic and selective motor control. Understanding the causal factors influencing function can help to better educate patients, families, and providers about realistic goals for treatments.

## Methods

Approval with a waiver of informed consent was obtained for this retrospective analysis through the Western IRB (study number 1249365).

### Causal model

A graphical causal model [31, 32] is proposed to explain how a set of commonly measured and potentially treatable neurological and orthopedic impairments affect function as measured by GMFM-66 (Fig 1 and Table 1). Model inputs are limited to those variables routinely collected during a motion analysis study for children with spastic CP at one center. The hypothesized relationships in the model, along with linearity assumptions, lead to conditional independence tests that assess the plausibility of the model and covariate adjustment sets for statistical models used to estimate the relative causal contributions of various factors [28].

The model is developed by considering the included clinically relevant variables and forming a hypothesis of how each variable influences every other variable in a causal relationship. If there is reasonable evidence or if a logical inference of *direct* causality exists, then graphically, a linkage is added from the causal variable to the effected variable. If a causal variable's effect is fully mediated by other included variables, then a direct effect is not included. The causal model proposed here starts with the neurologic injury that has led to the diagnosis of CP. The primary impairments resulting from this injury commonly manifest as spasticity and reductions in selective motor control, dynamic motor control, and strength. Each of these neurological factors is hypothesized to have a direct effect on gross motor function. Additionally, these primary impairments cause gait deviations through improper muscle activation patterns, which in turn impairs GMFM-66 by making intended movements more difficult. Selective voluntary motor control deficits also reduce strength by impeding the ability to isolate and volitionally activate muscles, resulting in functional weakness. Strength in turn affects both gait and gross motor function by altering and limiting movement strategies. The effect of age on gross motor function is accounted for both directly, reflecting motor learning associated with practice, and indirectly, reflecting neuromaturation and growth. Orthopedic deformities, in the form of contractures (Ankle, Knee, Hamstrings, and Hip) are caused, in part, by the improper muscle forces brought about by spasticity, poor strength, and poor motor control. These improper forces also influence the remodeling of bony torsions (femoral anteversion

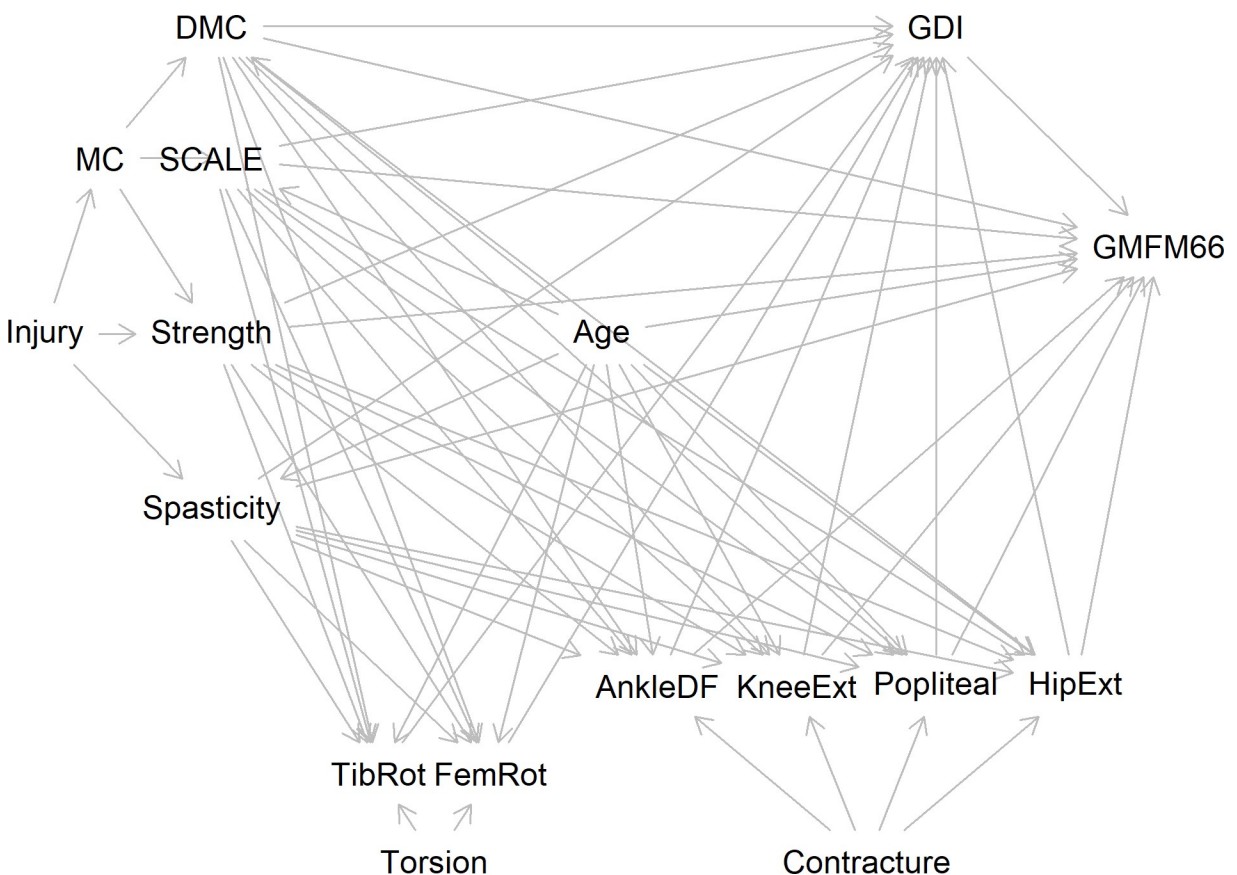

**Fig 1. Directed acyclic graph (DAG).** Alternatively called a causal Bayesian network. Representation of the causal model where arrows indicate hypothesized cause (tail) and effect (head) relationships. See Table 1 for hypothesized relationships of variables included.

and tibial torsion). Contractures and long-bone torsions both affect gait, and the contractures directly impede gross motor function by restricting movement. Note that we assume no direct effect of long bone torsion on gross motor function. There are undoubtedly other impairments associated with the injury that are not represented. For the present model, these other factors are ignored. This meaningful limitation and its impact on our results will be discussed later.

## Study data

We evaluated the model using 300 consecutive unique individuals with a diagnosis of CP who were seen at a single center for a quantitative 3D gait assessment including electromyography (EMG) of the lower extremity and had complete datasets. All studies were conducted by an experienced physical therapist. The center collecting the data was fully accredited during the times of data collection (Center for Motion Laboratory Accreditation, cmlainc.org). For all bilateral measures, averages between sides were used, including for unilaterally involved individuals. The following metrics categorized by ICF domains were used for the model.

**Body functions and structure.** *Spasticity* was computed on a 0–5 scale derived from multiple measures collected during physical examination [33]. The spasticity scale included modified Ashworth scales [34] recorded for between 3 to 5 muscles, Duncan-Ely tests of rectus femoris spasticity, beats of clonus, and ankle plantarflexor differences between initial catch of quick stretch and maximal values [33].

**Table 1. Causal model.**

| Clinical Feature | Measure | Causes |
|---|---|---|
| Injury | | |
| Age | Age | |
| Motor Control | MC, Latent | Injury |
| Selective Motor Control | SCALE | Motor Control, Age |
| Dynamic Motor Control | DMC | Motor Control, Age |
| Strength[a] | Strength | Injury, Motor Control, Age |
| Spasticity | Spasticity | Injury, Age |
| Contracture | Latent | |
| Ankle Dorsiflexion Contracture[b] | AnkleDF | Strength, Spasticity, Dynamic & Selective voluntary motor control, Age |
| Knee Extension Contracture[b] | KneeExt | Strength, Spasticity, Dynamic & Selective voluntary motor control, Age |
| Hamstrings Contracture[b] | Popliteal | Strength, Spasticity, Dynamic & Selective voluntary motor control, Age |
| Hip Extension Contracture[b] | HipExt | Strength, Spasticity, Dynamic & Selective voluntary motor control, Age |
| Torsion | Latent | |
| Tibial Torsion[c] | TibRot | Strength, Spasticity, Dynamic & Selective voluntary motor control, Age |
| Femoral Anteversion[c] | FemRot | Strength, Spasticity, Dynamic & Selective voluntary motor control, Age |
| Gait Pattern | GDI | Strength, Spasticity, Dynamic & Selective voluntary motor control, Age, Ankle ROM, Knee ROM, Popliteal ROM, Hip ROM, Tibial Torsion, Femoral Anteversion |
| Gross Motor Function | GMFM-66 | Strength, Spasticity, Dynamic & Selective voluntary motor control, Age, Ankle ROM, Knee ROM, Popliteal ROM, Hip ROM, Gait Pattern |

Features included in the causal model illustrated in Fig 1. Measures here match the Fig 1 variable names. The hypothesized direct causes of each feature are listed.

a. Because we measure strength in an age-adjusted manner, we do not include an explicit causal link in our graphical model used for testing plausibility and determining adjustment sets.

b. Ankle, Knee, Hamstrings, and Hip contractures are influenced by an unmeasured latent variable Contracture so that Contracture → Ankle Dorsiflexion Contracture, Contracture → Knee Extension Contracture, Contracture → Hamstrings Contracture, Contracture → Hip Extension Contracture.

c. Tibial Torsion and Femoral Anteversion are influenced by an unmeasured latent variable Torsion so that T → Tibial Torsion, T → Femoral Anteversion.

*Strength* was measured by both manual muscle testing grade and hand-held dynamometry of lower extremity muscle groups. Dynamometry values were expressed as z-scores relative to typically developing individuals after controlling for gender, age, and height [35, 36]. These were then combined with manual grades to yield a composite 0–5 strength scale.

*Selective motor control* was measured using the Selective Control Assessment of the Lower Extremity (SCALE). Total limb SCALE values (0–10 scale) were used as an overall representation of selective motor control [37].

Limb rotational deformities were measured with an inclinometer. *Femoral anteversion* was estimated from the midpoint between maximum inward and outward rotation of the hip with the subject prone and knee flexed to 90˚ [38]. *Tibial torsion* was estimated using bimalleolar and epicondyle landmarks with the subject supine and the knee extended [39, 40]. Typically developing values of 26.9˚ for femoral anteversion and 16.0˚ for tibial torsion were subtracted from raw data to reflect deviations from typical developing norms, with positive values internally directed [41].

Sagittal plane joint end ranges of motion were measured with handheld goniometry. Maximal *hip extension* was measured with the opposite hip flexed, *knee extension* in a supine position, *popliteal angle* with the opposite leg extended, and *ankle dorsiflexion* in a prone position with the knee at maximal extension [39, 40]. Joint contractures were expressed as the measured raw end range values offset by reported typically developing values: 0˚ for hip extension contracture, 4˚ of hyperextension (recurvatum) for knee extension contracture, 25.6˚ popliteal angle for hamstrings contracture, and 21.3˚ for ankle dorsiflexion contracture [41].

**Activity and participation.** *Gait quality* was measured by the gait deviation index (GDI), computed using published techniques [42]. The GDI is a validated measure of overall gait deviations, scaled so that 100 (10) is the mean (sd) for typically developing controls.

*Dynamic motor control* was expressed by the Walking Dynamic Motor Control Index (Walk-DMC), calculated from electromyography collected during walking at self-selected speed. Walk-DMC is a validated measure of motor control that is scaled so that 100 (10) is the mean (standard deviation) of typically developing peers. Walk-DMC values were computed based on the methods described by Steele *et al*. [43], using an aggregate analysis of 5–7 self-selected speed trials obtained as part of the clinical gait analysis protocol.

*Gross motor function classification level* (GMFCS) and the 66 item *gross motor function measure* (GMFM-66) were recorded by the physical therapist conducting the examination.

**Personal factors.** *Age* at assessment was considered to reflect maturation of strength and control. Height, weight, and gender were recorded and used to normalize dynamometry strength measures, but otherwise were not hypothesized to be explicit causal factors in this analysis.

**Health condition.** *CP subtype* (hemiplegia, diplegia, quadriplegia) was determined from the medical record.

Variables were transformed as follows. First, each variable was standardized using the sample mean and standard deviation. Next, the sign of the variable was adjusted so that larger values indicate less impairment and smaller values indicate more impairment. This was done so that positive effect sizes imply a higher score (better function) on the GMFM-66 scale. For age, larger values retain the conventional meaning of older.

## Analysis

Implied conditional independencies were identified to test the plausibility of the proposed model [31]. These conditional independencies are a list of partial correlations that must be zero or insignificant for the observed data to be consistent with the structure of the hypothesized causal model [44]. Meeting these conditions is necessary, but not sufficient, for a causal model to be correct, since there can be multiple models that meet the same implied independencies.

We chose to use linear models to estimate the magnitude and sign of total and direct causal effects for each factor of interest. A linear model was built for each factor whose effect size was being investigated using the adjustment set identified from the causal model. Adjustment sets are lists of covariates that, when included in the linear model, allow the causal effect of a particular factor to be estimated by controlling appropriately for other factors. Only main effects were allowed in the linear models (no interactions). This decision was driven by both causal and practical considerations. From a causal perspective, we did not see clear rationale for interaction effects, and, if they did exist, we assumed they would likely be substantially smaller than the main effects. From a practical perspective, we know that spurious significant interactions are likely, given the number of factors under consideration, and that approximately 16x more samples are needed to detect significant interactions compared to main effects [45]. Thus, in

light of our relatively coarse measurements and modest sample size, we settled on a main-effects-only model. In addition to deriving causal effect sizes for each *individual* factor, we also built a predictive model to examine the total causal effect of the factors taken together, enabling estimates of GMFM-66 as a function of the key clinical measures of neurological and orthopedic impairment.

We also derived bivariate linear models to assess the apparent, non-causal, effect sizes for each of the factors considered in the causal model. For example, we regressed GMFM-66 against Strength as a single predictor, then against Spasticity as a single predictor, and so on. The effect sizes from the bivariate models typify the estimates that result from the non-causal association analyses that pervade the literature. Thus, comparing the causal and bivariate results shows the interpretive errors that can arise from using a non-causal method.

A 10-fold cross-validation was performed to estimate the model's performance on out-of-sample observations, thereby quantifying the ability of model to predict GMFM-66. All modeling and statistics were performed with R (version 4.02, R Core Team, 2020). Causal model building and testing was done using dagitty [31].

## Results

Participants were typical for the population of CP patients referred for clinical gait analysis at our center (Table 2). We screened 314 individuals to reach the final cohort of 300 participants. Three subjects excluded had missing data due to visit time constraints while the other 11 were unable to complete testing due to an inability to follow directions. Of these 11, five were age six years or younger.

**Table 2. Participant characteristics.**

| Variable | Units | TD Value | GMFCS I | GMFCS II | GMFCS III |
|---|---|---|---|---|---|
| *mean (SD) unless noted* | | | | | |
| Number of Participants | | | 115 | 132 | 53 |
| CP Subtype (%) | | | | | |
| *Spastic diparesis* | | | 30 (26) | 95 (72) | 47 (89) |
| *Spastic hemiparesis* | | | 86 (74) | 32 (24) | 0 (0) |
| *Spastic quadriparesis* | | | 0 (0) | 5 (4) | 6 (11) |
| Gender = Male (%) | | | 52 (45) | 75 (57) | 34 (64) |
| Age | Yr. | | 10.5 (3.5) | 11.6 (3.5) | 10.9 (3.3) |
| Walk-DMC | None | 100 | 91 (10.0) | 80 (12.2) | 66 (9.8) |
| GDI | None | 100 | 82 (10.7) | 71 (9.4) | 63 (8.5) |
| SCALE | None | 10 | 6.8 (1.4) | 5.2 (1.4) | 3.0 (1.4) |
| Spasticity | None | 1 | 1.6 (0.9) | 2.1 (1.1) | 2.9 (1.1) |
| Strength | None | 5 | 4.6 (0.3) | 4.2 (0.5) | 3.5 (0.6) |
| GMFM-66 | None | 100 | 87 (6.2) | 73 (7.0) | 57 (4.6) |
| Ankle Dorsiflexion Contracture | Deg. | 0 | -17 (6.1) | -20 (8.1) | -20 (8.2) |
| Knee Extension Contracture | Deg. | 0 | 1 (4.9) | -2 (7.2) | -7 (8.8) |
| Hamstrings Contracture | Deg. | 0 | -13 (17.4) | -24 (18.7) | -31 (18.3) |
| Hip Extension Contracture | Deg. | 0 | -1 (2.7) | -4 (5.1) | -8 (7.6 |
| Femoral Anteversion Deviation | Deg. | 0 | -1 (24.1) | -4 (25.7) | -11 (24.9) |
| Tibial Torsion Deviation | Deg. | 0 | -1 (5.8) | -2 (6.7) | -3 (7.0) |

Note: Raw values reported. Contractures are negative values. For femoral anteversion and tibial torsion, positive values are inwardly directed. For modeling purposes, variables were transformed so that more positive values indicated less impairment and more negative values indicated more impairment.

Standardized coefficients were computed to assess effect sizes for the various neurological and orthopedic factors (Fig 2). Comparing effect sizes between the causal model and the bivariate regression models shows the important impact that causal modeling can have on effect sizes. Bivariate (non-causal) effect sizes overestimated the importance of factors by 0.18–0.48 compared to their causal counterparts, which corresponded to bivariate overestimations of more than double at a minimum to several fold.

All but two of the 11 independencies implied by the structure of our proposed causal model were statistically insignificant (no adjustment for multiple comparisons). There were significant correlations between hip extension contracture and tibial torsion (r = .18) and knee extension contracture and tibial torsion (r = .28), both conditional on age, DMC, SCALE, spasticity, and strength. This result demonstrates that the structure of the proposed causal model is largely consistent with the observed data, suggesting that the proposed model is plausible.

Coefficients for the direct linear model were estimated for all causal factors (Table 3). The cross validation analysis resulted in an out-of-sample $r^2$ = 0.75 and a mean absolute error of 4.9 points (Fig 3).

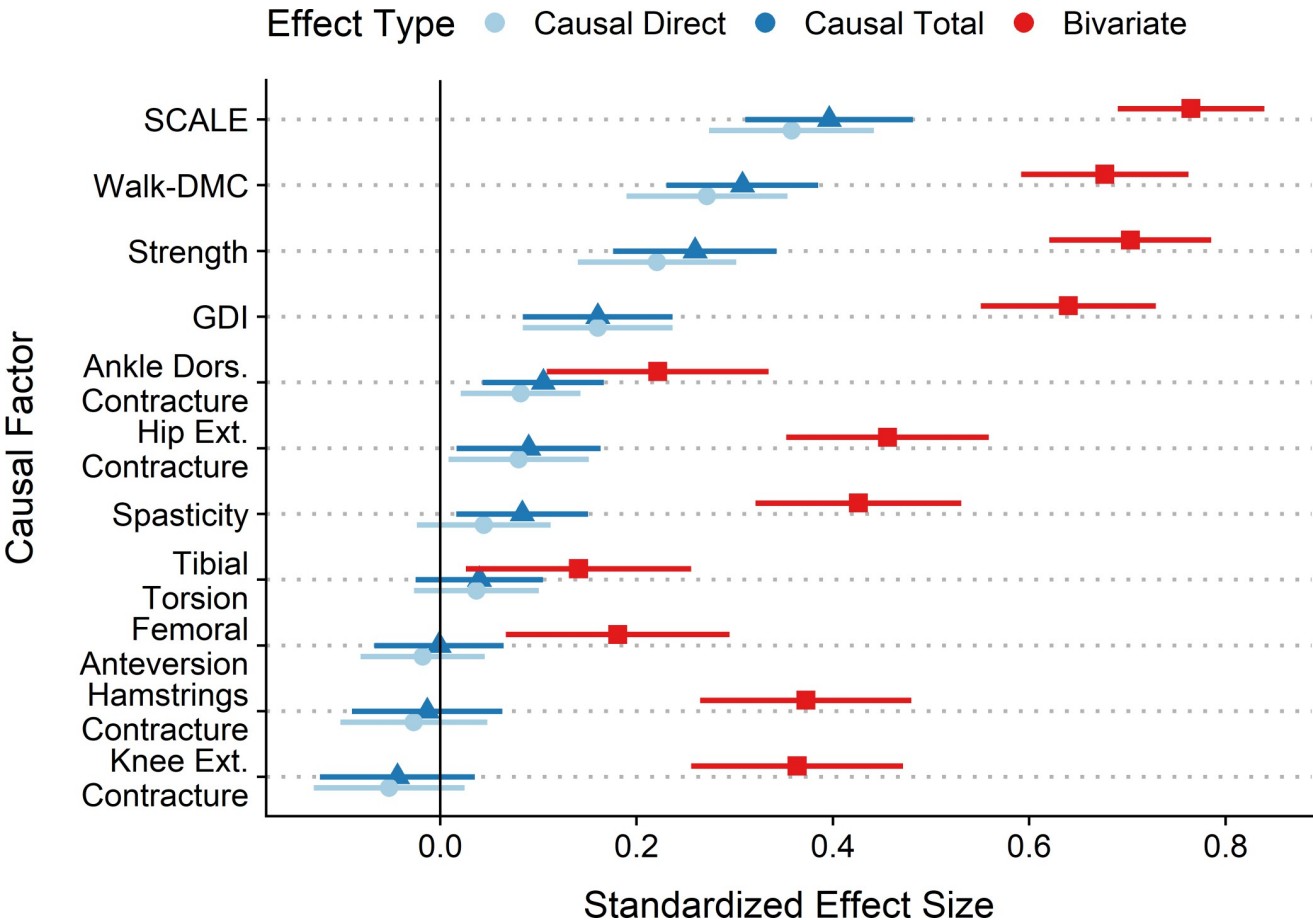

**Fig 2. Effect sizes.** Effect sizes are expressed as the standardized regression coefficients with 95% CI. **Causal Effect Sizes:** Motor control (SCALE and Walk-DMC) has the largest effects on GMFM-66, followed closely by Strength. Spasticity has a modest total effect, and a substantial decrease from total to direct effect size, suggesting its action is mediated by other factors. Ankle Dorsiflexion and Hip Extension contractures have effect sizes similar to Spasticity. Other orthopedic impairments do not meaningfully influence GMFM-66. Note that due to the hypothesized causal model structure, the direct and total effects of GDI are equal. **Bivariate Effect Sizes**: Comparison of causal effect sizes to bivariate shows the impact that causal modeling has on the estimated importance of clinical factors. The most obvious difference is one of magnitude–where bivariate effect sizes significantly overestimate the influence of each factor.

**Table 3. Predictive model coefficients.**

| Variable | β | SE | p | Units | Range | TD | Description |
|---|---|---|---|---|---|---|---|
| **Intercept** | 14.7 | 4.3 | < .01 | S | NA | NA | GMFM-66 linear intercept [7] |
| **Age** | 0.08 | 0.12 | .50 | Yr. | NA | NA | Child's age in decimal years (*e.g.* 10.4 yrs) |
| **SCALE** | 2.3 | .27 | < .01 | S | 0–10 | 10 | SCALE total limb score bilateral average (0–10) [37] |
| **Walk-DMC** | 0.24 | 0.04 | < .01 | S | 22–100+* | 100 (10) | Dynamic motor control scaled from normal [43] |
| **Strength** | 4.5 | 0.85 | < .01 | S | 0–5 | 5 | Multivariate strength scale [35] |
| **Spasticity** | -0.57 | 0.38 | .13 | S | 0–5 | 0 | Multivariate spasticity scale [33] |
| **GDI** | 0.17 | 0.04 | < .01 | S | 0–100+* | 100 (10) | Gait deviation index [42] |
| **Ankle Dors. Contracture** | -0.13 | 0.05 | < .01 | Deg. | > = 0 | 0 | Limitation in achieving > = 21.3˚ dorsiflexion with knee extended [39–41] |
| **Hip Ext. Contracture** | -0.17 | 0.08 | .02 | Deg. | > = 0 | 0 | Limitation in achieving > = neutral hip extension [39, 40] |
| **Knee Ext. Contracture** | -0.11 | 0.07 | .10 | Deg. | > = 0 | 0 | Limitation in achieving > = 4.0˚ knee extension with hip at neutral [39–41] |
| **Hamstrings Contracture** | -0.01 | 0.02 | .54 | Deg. | > = 0 | 0 | Limitation in achieving > = 25.6˚ knee extension with hip flexed to 90˚ (Popliteal Angle) [39–41] |
| **Femoral Anteversion** | -0.01 | 0.02 | .44 | Deg. | > = 0 | 0 | Difference from typical anteversion (26.9˚) [38, 41] |
| **Tibial Torsion** | -0.08 | 0.08 | .21 | Deg. | > = 0 | 0 | Non-weightbearing bimalleolar axis difference from typical (16.0˚) [39–41] |

S = unitless scale variable defined in description.

NA = not applicable, TD = typically developing value, SE = standard error, p = p-value.

*Both Walk-DMC and GDI are scaled Z-scores that can exceed the mean typical value of 100, additionally Walk-DMC has a theoretical minimum value dependent upon the scaling values.

*To obtain subject GMFM66 use equation*: $GMFM66 = \text{Intercept} + \Sigma_i \beta_i \cdot \text{Variable}_i$.

## Discussion

A causal model of function in children with CP provided estimates of direct and total effect sizes for key clinical impairments, and explained 75% of the variance in the GMFM-66 scores in an out-of-sample test set (10-fold cross-validation). The model is plausible and can be evaluated using measurements collected during routine gait analysis evaluations. Selective voluntary motor control, dynamic motor control, and strength had the largest total effects on GMFM-66. Gait impairments (GDI) had the next largest impact on GMFM-66. Note that GDI is not directly manipulable by treatment, but rather it changes in response to modification of orthopedic or neurological factors. Ankle and hip contractures had small effects on GMFM-66, while other orthopedic deformities had statistically insignificant effects. The causal effects of most factors were found to be substantially smaller than those obtained from bivariate analysis.

Our findings have important clinical implications. First, the effects of these measures on gross motor function can guide targets for treatment and inform expectations regarding possible improvements. Second, the results indicate that 75% of gross motor function as measured by the GMFM-66 can be accounted for by measures commonly recorded during clinical gait analysis. Finally, the results clearly show the errors that can arise from computing non-causal, bivariate relationships among clinical factors in children diagnosed with CP. The interconnectedness of neurological and musculoskeletal impairments, gait, and function in this population places a premium on employing analyses that respect the complexity of the clinical condition.

Selective voluntary motor control, as measured by the SCALE composite limb score, had the highest effect in the total effects model. The SCALE score has previously been associated with GMFM-66, and our analysis confirms that this association appears to be causal [46]. Dynamic motor control, as measured by Walk-DMC [43], also had a large effect size, even after controlling for selective voluntary motor control. This suggests that the neurological

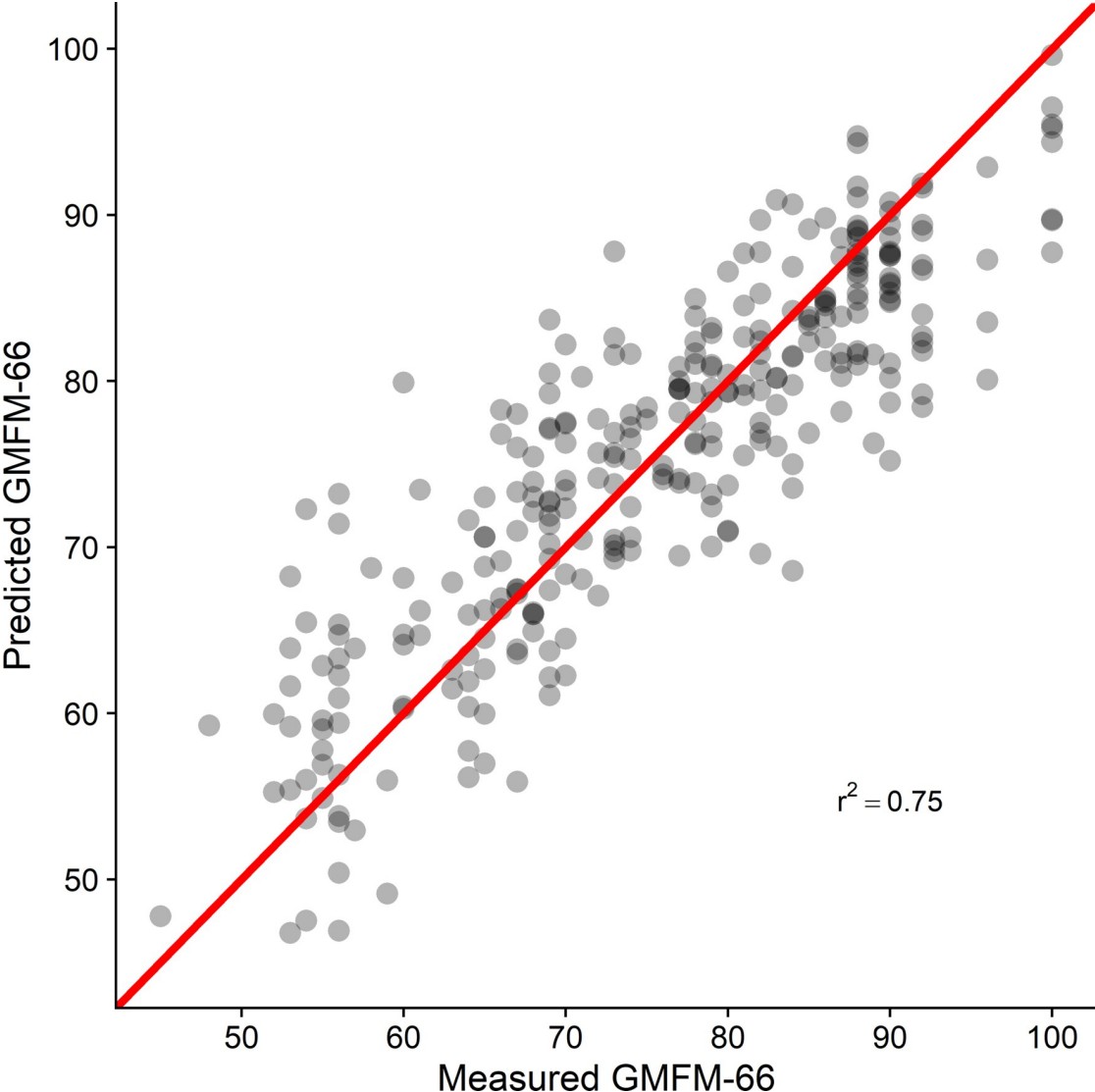

**Fig 3. Predicted *vs.* measured GMFM-66.** Data from a 10-fold cross validation of the final total effects linear model. The diagonal line indicates perfect agreement. The cross-validated $r^2 = 0.75$ and mean absolute error = 4.9 points.

pathways employed during quasi-static tasks and those used during gait are not the same, and both are important for function. In our proposed model, selective and dynamic motor control deficits share a common cause and are hence correlated (Walk-DMC ← Injury → SCALE). Walk-DMC has previously been associated with changes in gait measures after treatment, including overall gait deviation (GDI), consistent with the proposed causal diagram (Fig 1) in this study [47]. The same study demonstrated that Walk-DMC was associated with changes in the sports and physical function sub-score of the pediatric outcomes data collection instrument (PODCI). While the PODCI sports and physical function score is not identical to the GMFM-66, it is reasonable to think that there is a significant overlap in the underlying construct of the two measures. Selective and dynamic motor control and are difficult to alter with treatments or therapies. Nevertheless, our results suggest that we should continue efforts towards finding control-enhancing therapies, as these hold the greatest prospect for impacting overall function.

Spasticity is another common primary neurological impairment observed in children diagnosed with CP. Both the direct and total effects of spasticity were shown to be relatively small. Thus, intrathecal baclofen pump implantation, selective dorsal rhizotomy, or chemodenervation therapies aimed at spasticity reduction should not be expected to appreciably improve gross motor function. This prediction is consistent with what has been observed in outcome analyses of these treatments [16, 48, 49].

Age did not have a significant effect. Thus, independent of neurological and orthopedic impairments, age alone is not expected to alter gross motor function substantially. Contradicting this finding is previous work showing that GMFM-66 improves from an early age and begins to plateau around age 8 [50]. Approximately 85% of the individuals in this study were older than 7.5, and this may explain our finding that age was not an important cause of GMFM-66. In addition, the largest improvements due to ageing are observed in Gross Motor Function Classification (GMFCS) I individuals [51], who make up only 38% of our study population. Furthermore, we made the simplifying assumption of linearity, which cannot accurately model the growth and plateauing nature of previously observed GMFM-66 *vs.* age response. Lastly, the well-known GMFM-66 *vs.* age curves are stratified by GMFCS. While GMFCS *may* constitute a reasonable adjustment set, our causal model would suggest the nature of GMFM-66 changes are more complex. In fact, GMFCS level can be thought of as a causal consequence of many factors in our model. Put another way, children with poorer strength and motor control and worse spasticity are likely to be classified with a GMFCS level indicating greater severity of the disease. Thus, adjusting for GMFCS *alone* may not fully capture the effects that motor control, strength, orthopedic impairments, and gait impairments have on function. The lack of a direct age effect in our model suggests that changes in other factors with age (*e.g.* contractures, strength, gait quality) may explain some of the marginal age effect observed in the GMFM-66.

The effect sizes of strength and gait quality (GDI) were both significant, suggesting that they are important causes of function. Both variables can be improved by treatment (strengthening, physical therapy, orthopedic surgery). While our model suggests that improvement in these domains can raise gross motor function, the evidence for this in practice is not clear. Outcomes of these therapies have not consistently demonstrated improved function in patients [52]. One possible explanation could be the magnitude of strength or GDI improvement achieved with treatment. The standardized effect size for strength and GDI are 0.26 and 0.16, respectively. This means that for one standard deviation of strength improvement, we should expect 0.26 standard deviations of GMFM-66 improvement (3.2 points for the data in this study), and for one standard deviation of GDI improvement we should expect 0.16 standard deviations in GMFM-66 improvement (2.0 points). Typical GDI improvements following surgery are on the order of 7 points, which would correspond to ~1.3 point improvement on GMFM-66 scores, well below the reported minimally clinical important difference [53].

Similar to Kim and Park's less comprehensive causal model of function [30], we found that both spasticity and strength had significant effects on GMFM-66. However, the inclusion of additional causal factors significantly changed the absolute and relative magnitude of the effect sizes. Kim and Park estimated the effect sizes for spasticity and strength to be 0.34 and 0.45, respectively, while we found them to be 0.08 and 0.26. Differences in testing methods for these subjective measures may impact absolute effect sizes, but should have less impact on relative effect sizes, which were also substantially different in our model compared to Kim and Park's (4.82 and 3.25, respectively).

One limitation to this study was the exclusion of 14 individuals (4%) due to missing data. In some cases, data was not collected due to the patient's inability to follow instructions. These missing data have potential to bias the results since some of the lowest functioning individuals

may have been excluded. However, only six subjects (2%) were excluded for this reason. In addition, both sides of bilateral measures for unilateral individuals were included and averaged. While this may diminish the influence of the condition on the hemiparetic side, GMFM-66 is a score for an individual, allowing compensatory contributions of the non-hemiparetic side, and thus we reasoned that averaging sides was an appropriate approach. Finally, this is a retrospective analysis of cross sectional data and all causal relationships are hypothesized based on past observations or logistically based assumptions.

## Summary

A causal diagram is a hypothesis. Since counterfactual observations (*e.g.*, a clone of a specific child with one clinical parameter altered) do not exist, causal diagrams can never be fully proven or falsified. They can, however, be tested for plausibility from both a domain knowledge and statistical perspective. We showed that the proposed diagram is plausible and explained 75% of the variance in GMFM-66. Nevertheless, unobserved and unmeasured variables may play a significant role. As data were limited to variables retrospectively collected from standard gait analysis studies at a single center, some potentially important factors were omitted. Cognition may be one such omitted but important variable [54]. It is likely that cognition plays a meaningful role in function and including this variable could change the effect sizes of the factors we have examined. Socio-economic factors, access to resources, participation in the community, availability of recreational sports opportunities, emotional health, and myriad other complex factors could also be considered in future extensions of this model.

This study should be viewed as a starting point, and not as a definitive model of causality and function in children with CP. We proposed a model limited to data available from existing motion analysis studies, demonstrated its plausibility, and estimated the causal effect sizes of key clinical factors. We also showed that important differences in interpretation arise when causality, rather than association, is considered. The strength of our approach is the explicit proposal of a causal diagram, leading to *a priori* statistical models. This contrasts with much of the previous work in this area, which commonly examines *ad hoc* bivariate relationships in an unstructured methodology. In many cases, the bivariate approach leads to significant and strong correlations, but these findings may be misleading since they lack the proper causal context.

As noted earlier, lurking behind virtually every study of association is a hidden belief in a causal relationship. Except for purely predictive models, we suspect no researchers examine associations without some underlying hypothesis about causality. While our proposed model is imperfect, it is transparent and explicit. Additional work is needed to discover more complete causal models and to devise experiments that confirm or refute the results of these models. The complexity of CP and the relationships among commonly measured variables demand that causality be considered when interpreting data.

## Conclusions

Our model results highlight the challenges faced for improving function in children with CP. The largest effect sizes are found in the selective and dynamic motor control measures; impairments for which there are currently no highly effective treatments. In contrast, spasticity, which we can treat via rhizotomy or baclofen pump, has a very small impact on gross motor function. Strength and gait quality both have significant, but modest, effect sizes. However, improvements in these domains following treatment tend to be small, leading to minimal functional gains. If we hope to improve function through these domains, we need more effective treatments for strength and gait.

## Supporting information

**S1 Data.**
(CSV)

## Author Contributions

**Conceptualization:** Bruce A. MacWilliams, Michael H. Schwartz.

**Data curation:** Bruce A. MacWilliams, Sarada Prasad, Amy L. Shuckra, Michael H. Schwartz.

**Formal analysis:** Bruce A. MacWilliams, Michael H. Schwartz.

**Investigation:** Bruce A. MacWilliams, Michael H. Schwartz.

**Methodology:** Bruce A. MacWilliams, Michael H. Schwartz.

**Project administration:** Bruce A. MacWilliams.

**Software:** Bruce A. MacWilliams, Michael H. Schwartz.

**Supervision:** Bruce A. MacWilliams, Michael H. Schwartz.

**Validation:** Bruce A. MacWilliams, Michael H. Schwartz.

**Visualization:** Bruce A. MacWilliams, Michael H. Schwartz.

**Writing – original draft:** Bruce A. MacWilliams, Michael H. Schwartz.

**Writing – review & editing:** Bruce A. MacWilliams, Sarada Prasad, Amy L. Shuckra, Michael H. Schwartz.

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
