## [Decision Letter · Decision Letter 0]

28 Feb 2022

PONE-D-21-29034

Causal factors affecting gross motor function in children diagnosed with cerebral palsy

PLOS ONE

Dear Dr. MacWilliams,

Thank you for submitting your manuscript to PLOS ONE. After careful consideration, we feel that it has merit but does not fully meet PLOS ONE’s publication criteria as it currently stands. Therefore, we invite you to submit a revised version of the manuscript that addresses the points raised during the review process.

We look forward to receiving your revised manuscript.

Kind regards,

YunJu Lee

Academic Editor

PLOS ONE

https://journals.plos.org/plosone/s/fileid=ba62/PLOSOne_formatting_sample_title_authors_affiliations.pdf".

2. As you are reporting a retrospective study of medical records, please ensure that you have discussed whether all data were fully anonymized before you accessed them and/or whether the IRB or ethics committee waived the requirement for informed consent. If patients provided informed written consent to have data from their medical records used in research, please include this information.

3. "We noted in your submission details that a portion of your manuscript may have been presented or published elsewhere. [DETAILS AS NEEDED] Please clarify whether this [conference proceeding or publication] was peer-reviewed and formally published. If this work was previously peer-reviewed and published, in the cover letter please provide the reason that this work does not constitute dual publication and should be included in the current manuscript.

Reviewers' comments:

Reviewer's Responses to Questions

**Comments to the Author**

1. Is the manuscript technically sound, and do the data support the conclusions?

Reviewer #1: Yes

Reviewer #2: Partly

2. Has the statistical analysis been performed appropriately and rigorously? 

Reviewer #1: Yes

Reviewer #2: Yes

3. Have the authors made all data underlying the findings in their manuscript fully available?

Reviewer #1: No

Reviewer #2: Yes

4. Is the manuscript presented in an intelligible fashion and written in standard English?

Reviewer #1: Yes

Reviewer #2: Yes

5. Review Comments to the Author

Reviewer #1: Thank you for asking me to review this interesting paper I like the idea behind this article. CP is a complex condition and there is a place for more complex analysis when evaluating treatments and training. The authors have done a great job in analysing and presenting the data and I believe that this paper could be important for researchers and clinicians working with people with cerebral palsy.

I want to start by pointing out that I am a clinician first and not confident yet in my role as a reviewer. My knowledge of the advanced statistical analysis method in this paper is limited. I have therefore asked my co-writer and docent Wim Grooten for help in this regard. We have also discussed other aspects of the manuscript, but all information was handled confidentially, and no information has been spread elsewhere. After discussing with him, we agreed that the statistical analysis was sound and the results interesting.

With that said, I have some minor questions and concerns after reading the manuscript.

Aim:

The aim of the study has not explicitly been formulated, not in the end of the introduction or in the abstract. I think to increase the readability of the paper, the authors could formulate their research questions and purpose more clearly on these places.

Introduction:

You use the words “effectively improve function” (in the first paragraph of the introduction) and “alter gross motor function substantially” (in the last paragraph on page 21). This is a little confusing for the reader and could use some clarification, what do you mean when you say effectively and substantially?

Results

On page 17, the second paragraph starts a bit strange. Does all the text belong to figure 2 or is it main text in the result section? The bold writing indicates that it belongs to the figure, but it is a little unclear.

Results/ discussion:

In the first paragraph on page 20 you state that contractures had a small effect on GMFM- 66. GMFM- 66 measures peak performance for the day and time when it was performed and does not evaluate quality of movement or endurance. So maybe contractures have little effect on GMFM- 66, but what about in everyday life or during walks? Do they have a bigger influence on gross motor function then?

Second paragraph on page 20 you state that “factors explaining 75% of gross motor function can be measured during routine gait analysis”. This might be true for clinicians that perform advanced gait analyses, but is this statement valid for all kinds of gait analyses? Furthermore, do you mean 75% of gross motor function or 75% of gross motor function measured with GMFM-66? If I understand you correctly it is 75% of the factors reflected in the GMFM- 66.

It was interesting to read that selective and dynamic motor control had the largest effect sizes in the model. I think that it is hard to differentiate gross motor control/ function from selective and dynamic motor control; if one of them improves, the others are bound to be affected. For what is gross motor skills if not variants of selective and dynamic movements? This could explain part of the result and could be worth mentioning in the discussion section.

On page 21 and 22 you discuss the influence of age on the results. The last sentence on page 21 I agree with but the start on page 22 is somewhat confusing for me. The sentence when you introduce GMFCS is hard to read because of the parentheses, I realise they must be there but maybe try rephrasing it. Overall, the first paragraph on page 22 is a bit hard to understand. If the reader is familiar with the gross motor curves and their plateaus it makes it a bit clearer, but it is still hard to follow. So maybe look it over one more time to see if it can be made more understandable.

I think the discussion is somewhat long, especially the part with the heading “summary”. This “summary” of two pages is, in my regard, not a summary, but rather a methodological consideration. However, most of topics are interesting but could be condensed under a different heading. Also, could the authors provide a “conclusion” in the end of the discussion? The conclusion available in the abstract could be used as a starting point here and, if possible, should reflect both the implication for clinical work and future research.

Reviewer #2: Thank you for submitting this manuscript aiming to overcome the bottleneck in enhance motor function in children with cerebral palsy. This is an important start for clinical practitioners to re-consider and implement efficacious intervention to support this population. I still have few suggestions and questions regarding the content and result interpretation.

1. The proposed causal model was based on clinical speculations regarding relationship between impairments and associated motor function and is somehow supported by the result of linear models. However, consider the nature of retrospective approach with cross-sectional data set, it still needs to be more conservative in claiming a cause-and-effect relationship, which may be more complicated than it is interpreted in the present manuscript. I still suggested to tone down the conclusions while adding more discussion regarding the potential bidirectional or mediation effect.

- For example, whether spasticity might be the mediating variables between selective motor control and functional outcome although the effect size of spasticity is small. This may also lead to a different judgement, together with the current evidence, on whether common tone management approaches are still with their values.

- “The effect of age on gross motor function is accounted for both directly, reflecting motor learning associated with practice, and indirectly, reflecting neuromaturation and growth.” This is an example of not knowing the cause-and-effect relationship or uncertain direction of the influence of one factor on the other. For example, age can have both the positive and negative impact on gait and functional performance. The negative influence may come from increased body weight and severer secondary impairments such as contracture. The other example is bony torsion, which may be worsen by on developing bone with walking.

2. Introduction: I would also suggest introducing CP (the first few sentences of current 2nd paragraph) before getting into more details regarding the evidence of current intervention effects. Please also add more information regarding the effect in treating different impairments in children with CP in the introduction or discussion

3. Introduction: Functional training and environmental modifications have showed stronger evidence in improving motor function in children with CP. However, there is insufficient review on relevant evidence in the introduction. Thus, it is suggested to add this information and discuss how it may help to support your findings or vice versa

4. Introduction: The following sentence is suggested to be removed

“ This assumption is entirely plausible. In fact, we demonstrate below that strength is an important causal factor, though with a substantially smaller impact than bivariate estimates would suggest. “

5. Method: Are there references to support the proposed causal model? There is no reference in the whole section of “Causal model”

6. Method: Although addressed in the discussion, I still have concerned toward averaging bilateral data in children with unilateral involvement. The effect of independent variables may be offset with it. This may be worth to first test which side of the data shows stronger impact on the functional outcome or the difference between affected and non-affected side plays a better role. In addition, what is the proportion of children with hemiplegia?

6. PLOS authors have the option to publish the peer review history of their article (what does this mean?). If published, this will include your full peer review and any attached files.

Reviewer #1: **Yes: **Emma Nylén, Registered Physiotherapist, Master of Science

Reviewer #2: No

---

## [Author Response · Author response to Decision Letter 0]

18 Apr 2022

These are contained in a separate document but here they are ...

Response to Reviewers:

We would like to thank the reviewers for their comments, which we address point by point below. Additions or changes to the text from the original manuscript are recorded per journal instructions using the track changes tool in Word. 

Reviewer’s Comments:

Reviewer 1

Thank you for asking me to review this interesting paper I like the idea behind this article. CP is a complex condition and there is a place for more complex analysis when evaluating treatments and training. The authors have done a great job in analysing and presenting the data and I believe that this paper could be important for researchers and clinicians working with people with cerebral palsy.

I want to start by pointing out that I am a clinician first and not confident yet in my role as a reviewer. My knowledge of the advanced statistical analysis method in this paper is limited. I have therefore asked my co-writer and docent Wim Grooten for help in this regard. We have also discussed other aspects of the manuscript, but all information was handled confidentially, and no information has been spread elsewhere. After discussing with him, we agreed that the statistical analysis was sound and the results interesting.

With that said, I have some minor questions and concerns after reading the manuscript.

Aim:

The aim of the study has not explicitly been formulated, not in the end of the introduction or in the abstract. I think to increase the readability of the paper, the authors could formulate their research questions and purpose more clearly on these places.

Response: Aim and purpose are given in the abstract (Aim paragraph), and at the end of the introduction: “In this study, we hypothesize and test a causal model for GMFM-66 that examines the influence of key neurological and orthopedic impairments commonly measured during a gait analysis visit, which includes a comprehensive physical exam, standard functional measures, and 3D kinematics and electromyography collected during walking.”

Revision: Altered the aim sentence at the end of the introduction to read: “The aim of this study is to hypothesize and test a causal model for GMFM-66 that examines the influence of key neurological and orthopedic impairments commonly measured during a gait analysis visit, which includes a comprehensive physical exam, standard functional measures, and 3D kinematics and electromyography collected during walking.” 

Introduction:

You use the words “effectively improve function” (in the first paragraph of the introduction) and “alter gross motor function substantially” (in the last paragraph on page 21). This is a little confusing for the reader and could use some clarification, what do you mean when you say effectively and substantially?

Response: Both adjectives were meant to describe that only small responses occurred or would be expected.

Revision: In the first instance we remove “effectively” so that the sentence now reads: “There is little evidence that common treatments for the CP child improve function.” In the second instance we have not revised as we state first that Age did not have a significant effect in the model. With maturation it is reasonable to expect some improvement in function, at least in very young children, as motor learning increases. Yet the model did not find a significant effect of Age. This is why we state: “…age alone is not expected to alter gross motor function substantially.” Introduction has been expanded and reorganized.

Results

On page 17, the second paragraph starts a bit strange. Does all the text belong to figure 2 or is it main text in the result section? The bold writing indicates that it belongs to the figure, but it is a little unclear.

Response: All text does belong to Figure 2. Hopefully with the manuscript properly laid out this will alleviate the confusion. For the revision we have inserted labels to mark these caption paragraphs as well as line separators to isolate the caption text. 

Results/ discussion:

In the first paragraph on page 20 you state that contractures had a small effect on GMFM- 66. GMFM- 66 measures peak performance for the day and time when it was performed and does not evaluate quality of movement or endurance. So maybe contractures have little effect on GMFM- 66, but what about in everyday life or during walks? Do they have a bigger influence on gross motor function then?

Response: We only measure function via GMFM-66 and so we cannot extend these results to other measures of function. The influence of contractures on gait as measured by the gait deviation index is accounted for as shown in Figure 1. 

Second paragraph on page 20 you state that “factors explaining 75% of gross motor function can be measured during routine gait analysis”. This might be true for clinicians that perform advanced gait analyses, but is this statement valid for all kinds of gait analyses? Furthermore, do you mean 75% of gross motor function or 75% of gross motor function measured with GMFM-66? If I understand you correctly it is 75% of the factors reflected in the GMFM- 66.

Response: The reference to routine gait was meant to reflect our own practice, which in our experience as an accredited laboratory, is generally in line with other accredited labs. Gross motor function measures here always refer the GMFM-66 measure. This is further clarified in the Methods: “Model inputs are limited to those variables routinely collected during a motion analysis study for children with spastic CP at one center.”

Revision: Altered the referenced sentence to now read: “Second, the results indicate that 75% of gross motor function as measured by the GMFM-66 can be accounted for by measures commonly recorded during clinical gait analysis.”

It was interesting to read that selective and dynamic motor control had the largest effect sizes in the model. I think that it is hard to differentiate gross motor control/ function from selective and dynamic motor control; if one of them improves, the others are bound to be affected. For what is gross motor skills if not variants of selective and dynamic movements? This could explain part of the result and could be worth mentioning in the discussion section.

Response: This is precisely true, but equally true are the complex relationships that exist between motor control and function as illustrated in Figure 1. As shown in Figure 2, simple bivariate relationships to GMFM-66 would overestimate motor control measure contributions by more than double what we have found in the proposed model. Finally, perhaps contradicting somewhat the comment above, we found these two measures actually have a large degree of independency as we explain: 

“Dynamic motor control, as measured by Walk-DMC (39), also had a large effect size, even after controlling for selective voluntary motor control. This suggests that the neurological pathways employed during quasi-static tasks and those used during gait are not the same, and both are important for function. In our proposed model, selective and dynamic motor control deficits share a common cause and are hence correlated (Walk-DMC ← Injury → SCALE).”

On page 21 and 22 you discuss the influence of age on the results. The last sentence on page 21 I agree with but the start on page 22 is somewhat confusing for me. The sentence when you introduce GMFCS is hard to read because of the parentheses, I realise they must be there but maybe try rephrasing it. Overall, the first paragraph on page 22 is a bit hard to understand. If the reader is familiar with the gross motor curves and their plateaus it makes it a bit clearer, but it is still hard to follow. So maybe look it over one more time to see if it can be made more understandable.

Revision: Moved the citation so that parentheses are not juxtaposed. 

I think the discussion is somewhat long, especially the part with the heading “summary”. This “summary” of two pages is, in my regard, not a summary, but rather a methodological consideration. However, most of topics are interesting but could be condensed under a different heading. Also, could the authors provide a “conclusion” in the end of the discussion? The conclusion available in the abstract could be used as a starting point here and, if possible, should reflect both the implication for clinical work and future research.

Revision: We have rearranged the Summary section to include an ending Conclusions paragraph. 

Reviewer #2: Thank you for submitting this manuscript aiming to overcome the bottleneck in enhance motor function in children with cerebral palsy. This is an important start for clinical practitioners to re-consider and implement efficacious intervention to support this population. I still have few suggestions and questions regarding the content and result interpretation.

1. The proposed causal model was based on clinical speculations regarding relationship between impairments and associated motor function and is somehow supported by the result of linear models. However, consider the nature of retrospective approach with cross-sectional data set, it still needs to be more conservative in claiming a cause-and-effect relationship, which may be more complicated than it is interpreted in the present manuscript. I still suggested to tone down the conclusions while adding more discussion regarding the potential bidirectional or mediation effect.

- For example, whether spasticity might be the mediating variables between selective motor control and functional outcome although the effect size of spasticity is small. This may also lead to a different judgement, together with the current evidence, on whether common tone management approaches are still with their values.

- “The effect of age on gross motor function is accounted for both directly, reflecting motor learning associated with practice, and indirectly, reflecting neuromaturation and growth.” This is an example of not knowing the cause-and-effect relationship or uncertain direction of the influence of one factor on the other. For example, age can have both the positive and negative impact on gait and functional performance. The negative influence may come from increased body weight and severer secondary impairments such as contracture. The other example is bony torsion, which may be worsen by on developing bone with walking.

Response: Regarding spasticity both direct and indirect (mediating) effects are small, so we can confidently say that alteration in spasticity will not have large effect on GMFM-66. This has been supported in a recent publication where subjects matched at an early baseline age and studied as adults showed equivalent changes in GMFM-66 independent of spasticity reduction status. We agree with the points about directionality of factors however this is inherent in the data (some positive changes, some negative changes) and the cause-effect considerations as shown in Figure 1. For example age is causally related to all contractures (likely that contractures increase with age as noted) and is causally related to GMFM-66 (we know that for age <8 GMFM-66 should improve with age). Finally, we have tried to be extremely careful and forward about the limitations of the model including: 

“A causal diagram is a hypothesis. Since counterfactual observations (e.g., a clone of a specific child with one clinical parameter altered) do not exist, causal diagrams can never be fully proven or falsified.”

“As data were limited to variables retrospectively collected from standard gait analysis studies at a single center, some potentially important factors were omitted.“

“This study should be viewed as a starting point, and not as a definitive model of causality and function in children with CP. We proposed a model limited to data available from existing motion analysis studies …”

“While our proposed model is imperfect, it is transparent and explicit. Additional work is needed to discover more complete causal models and to devise experiments that confirm or refute the results of these models.”

Revision: Added the following sentence to the limitations: “Finally, this is a retrospective analysis of cross sectional data and all causal relationships are hypothesized based on past observations or logistically based assumptions.”

2. Introduction: I would also suggest introducing CP (the first few sentences of current 2nd paragraph) before getting into more details regarding the evidence of current intervention effects. Please also add more information regarding the effect in treating different impairments in children with CP in the introduction or discussion

Revision: Paragraphs added to introduction to expand definition and incidence of CP and treatment modalities. Introduction reorganized to include further information of common treatment modalities.

3. Introduction: Functional training and environmental modifications have showed stronger evidence in improving motor function in children with CP. However, there is insufficient review on relevant evidence in the introduction. Thus, it is suggested to add this information and discuss how it may help to support your findings or vice versa

Response: It is of course difficult to give a comprehensive review of the literature regarding treatment modalities and their efficacy in CP. We have attempted to limit the discussion to those studies that have directly measured GMFM-66, the outcome variable that was studied in the model. 

Revision: We have reorganized and added information to the introduction, however, we have been unable to find any evidence that functional training and environmental modifications have shown long-term improvements in GMFM-66. We did find one additional citation (Ketelaar et al., 2001) regarding short-term improvement and have added that. 

4. Introduction: The following sentence is suggested to be removed

“ This assumption is entirely plausible. In fact, we demonstrate below that strength is an important causal factor, though with a substantially smaller impact than bivariate estimates would suggest. “

Response: The authors feel that this is a key point to justify and promote the use of explicit causal inferences and modeling. Thus we have retained some of this argument. 

Revision: The first sentence has been deleted and the second modified. 

5. Method: Are there references to support the proposed causal model? There is no reference in the whole section of “Causal model”

Response: Two citations were included to support this section: 

Pearl J. Causal diagrams for empirical research. Biometrika. 1995 Dec 1;82(4):669–88.

Textor J, van der Zander B, Gilthorpe MS, Liśkiewicz M, Ellison GT. Robust causal inference using directed acyclic graphs: the R package ‘dagitty.’ Int J Epidemiol. 2016 Dec 1;45(6):1887–94.

Revision: A more general citation of causal modelling has been added to support this paragraph: Pearl J. Causality [Internet]. 2nd ed. Cambridge: Cambridge University Press; 2009 

6. Method: Although addressed in the discussion, I still have concerned toward averaging bilateral data in children with unilateral involvement. The effect of independent variables may be offset with it. This may be worth to first test which side of the data shows stronger impact on the functional outcome or the difference between affected and non-affected side plays a better role. In addition, what is the proportion of children with hemiplegia?

Response: Early work not reported here compared results using: 

1) A random side of individuals with bilateral CP and the hemiparetic side of individuals with unilateral 

2) Side with best dynamic motor control of individuals with bilateral CP and the hemiparetic side of individuals with unilateral

3) Side with worst dynamic motor control of individuals with bilateral CP and the hemiparetic side of individuals with unilateral

4) Average used here

There was very little difference in the results based on these choices and logistically, the average values made the most sense to us given the nature of the specific motor tasks used for GMFM-66. The number of hemiplegic individuals is included in Table 2. There were 118 of the 300 or 39%.

---

## [Decision Letter · Decision Letter 1]

6 Jun 2022

Causal factors affecting gross motor function in children diagnosed with cerebral palsy

PONE-D-21-29034R1

Dear Dr. MacWilliams,

We’re pleased to inform you that your manuscript has been judged scientifically suitable for publication and will be formally accepted for publication once it meets all outstanding technical requirements.

Kind regards,

YunJu Lee

Academic Editor

PLOS ONE

Additional Editor Comments (optional):

Reviewers' comments:

Reviewer's Responses to Questions

**Comments to the Author**

1. If the authors have adequately addressed your comments raised in a previous round of review and you feel that this manuscript is now acceptable for publication, you may indicate that here to bypass the “Comments to the Author” section, enter your conflict of interest statement in the “Confidential to Editor” section, and submit your "Accept" recommendation.

Reviewer #1: All comments have been addressed

2. Is the manuscript technically sound, and do the data support the conclusions?

Reviewer #1: Yes

3. Has the statistical analysis been performed appropriately and rigorously? 

Reviewer #1: I Don't Know

4. Have the authors made all data underlying the findings in their manuscript fully available?

Reviewer #1: Yes

5. Is the manuscript presented in an intelligible fashion and written in standard English?

Reviewer #1: Yes

6. Review Comments to the Author

Reviewer #1: The authors have responded to the comments. And I feel satisfied with the answers and revisions.

As stated before my knowledge of advanced statistical analysis is limited therefore I can not give a more sound answer to that question.

7. PLOS authors have the option to publish the peer review history of their article (what does this mean?). If published, this will include your full peer review and any attached files.

Reviewer #1: No

---

## [Editor Report · Acceptance letter]

7 Jul 2022

PONE-D-21-29034R1 

Causal factors affecting gross motor function in children diagnosed with cerebral palsy 

Dear Dr. MacWilliams:

I'm pleased to inform you that your manuscript has been deemed suitable for publication in PLOS ONE. Congratulations! Your manuscript is now with our production department. 

Kind regards, 

on behalf of

Dr. YunJu Lee 

Academic Editor

PLOS ONE